# Ferroptosis and Senescence: A Systematic Review

**DOI:** 10.3390/ijms24043658

**Published:** 2023-02-11

**Authors:** Donatella Coradduzza, Antonella Congiargiu, Zhichao Chen, Angelo Zinellu, Ciriaco Carru, Serenella Medici

**Affiliations:** 1Department of Biomedical Sciences, University of Sassari, 07100 Sassari, Italy; 2Control Quality Unit, Azienda-Ospedaliera Universitaria (AOU), 07100 Sassari, Italy; 3Department of Chemical, Physical, Mathematical and Natural Sciences, University of Sassari, 07100 Sassari, Italy

**Keywords:** ferroptosis, senescence, aging, disease

## Abstract

Senescence is a cellular aging process in all multicellular organisms. It is characterized by a decay in cellular functions and proliferation, resulting in increased cellular damage and death. This condition plays an essential role in the aging process and significantly contributes to the development of age-related complications. On the other hand, ferroptosis is a systemic cell death pathway characterized by excessive iron accumulation followed by the generation of reactive oxygen species (ROS). Oxidative stress is a common trigger of this condition and may be induced by various factors such as toxins, drugs, and inflammation. Ferroptosis is linked to numerous disorders, including cardiovascular disease, neurodegeneration, and cancer. Senescence is believed to contribute to the decay in tissue and organ functions occurring with aging. It has also been linked to the development of age-related pathologies, such as cardiovascular diseases, diabetes, and cancer. In particular, senescent cells have been shown to produce inflammatory cytokines and other pro-inflammatory molecules that can contribute to these conditions. In turn, ferroptosis has been linked to the development of various health disorders, including neurodegeneration, cardiovascular disease, and cancer. Ferroptosis is known to play a role in the development of these pathologies by promoting the death of damaged or diseased cells and contributing to the inflammation often associated. Both senescence and ferroptosis are complex pathways that are still not fully understood. Further research is needed to thoroughly investigate the role of these processes in aging and disease, and to identify potential interventions to target such processes in order to prevent or treat age-related conditions. This systematic review aims to assess the potential mechanisms underlying the link connecting senescence, ferroptosis, aging, and disease, and whether they can be exploited to block or limit the decay of the physiological functions in elderly people for a healthy longevity.

## 1. Introduction

Ferroptosis was first reported in 2012 [1,2], as an iron-dependent mechanism of non-apoptotic cell death [3], which has attracted the researchers’ interest due to its implications in the pathogenesis of various age-related disorders. Moreover, it can play both a negative and a positive role, as it may be involved in cancer initiation and development, as well as suppression, see for instance the review [4].

This iron-dependent regulated necrosis cell death is characterized by massive membrane damage mediated by uncontrolled lipid peroxidation [5], which is in turn triggered by redox mechanisms connected to the Fenton’s reaction. The latter is a rather common process in cells, involving the Fe(II/III)-catalysed oxidation of organic substrates by H_2_O_2_ through a multi-step formation of highly reactive oxidizing species, such as the hydroxyl and hydroperoxyl radicals (shortly summarized in Figure 1), that in some cases may involve also lipids and phospholipids at their carbon-carbon double bonds, especially the polyunsaturated fatty acids (PUFAs) of the plasma membrane or membrane-enclosed organelles.

Ferroptosis demonstrated to play a key role in cellular homeostasis leading to a gradual and progressive functional decay that eventually causes inevitable death. It was first identified in the early 2000s, but the understanding of the mechanisms was completed between 2012 and 2014, and ferroptosis was included among the 12 different types of cell death pathways, mechanistically distinct from apoptosis and other forms of necrosis, including necroptosis and pyroptosis [6,7].

Dixon and Stockwell (2019) provided an overview of the hallmarks of ferroptosis. It was Dixon’s research that demonstrated ferroptosis can occur in cells without a functioning mitochondrial electron transport chain (ETC), as seen in HT1080 cells [8].

The authors described the distinguishing features of ferroptosis as being related to three characteristics:(a)availability of redox-active iron(b)availability of chemically competent substrates to undergo peroxidation (e.g., PUFA phospholipids)(c)impairment of the cellular lipid repair system that normally prevents the accumulation of these lethal species.

Ferroptosis, as described by the authors, varies among different cell types and even within the same cell type, linked to a unique expression pattern for proteins involved in iron, lipid, and antioxidant processes.

Thus, the mechanisms governing ferroptosis are slowly being unveiled [9], but they seem to be mainly centred on cysteine and glutathione (GSH) metabolism and the ability of the phospholipid peroxidase GPX4 to prevent iron-dependent accumulation of peroxidised lipids. Basically, it is characterized by the failure of glutathione-dependent antioxidant defences, resulting in iron-catalysed peroxidation of polyunsaturated fatty acid phospholipids, which is countered by GSH activity as a ligand for cytosolic Fe^2+^ [10] and as a substrate for glutathione peroxidase-4, to eliminate membrane lipid peroxides leading, when accumulated, to ferroptotic cell death [8,11,12].

Many researchers have hypothesized that an increase in Fe^2+^ levels catalyses both the Fenton’s reaction cascade to generate hydroxyl radicals from hydrogen peroxide, and the consequent depletion of GSH, which in turn fuels the catalytic increase in lipid peroxidation, predisposing cells to ferroptosis and biological aging. Studies have shown the propagation of ferroptosis in a paracrine manner outward from the affected cells. Although the details of this mechanism are still unclear, the relatively stable and diffusible toxic lipid peroxidation end-products 4-hydroxynonenal and malondialdehyde may mediate this effect by contributing to cellular senescence [13]. Directly or indirectly induced senescence is proposed to be a gradual deterioration of cellular tissue and living organism functional characteristics associated with redox imbalances with chronic induction and upregulation of pro-inflammatory mediators (e.g., TNF-α, IL-1β, IL-6, COX-2, iNOS) and signalling pathways such as NF-KB, and subsequent reduction in the activity of antioxidant pathways [14].

During senescence, growth stops in a stable manner to limit the replication of old and damaged cells. Cells undergo morphological changes, chromatin remodelling and metabolic reprogramming by secreting predominantly proinflammatory factors called senescence-associated secretory phenotype (SASP) [15]. While aging, the amount of circulating iron is reduced, and tissue and intracellular iron stores increase, leading to deleterious effects on cellular functions due to redox imbalances causing ferroptosis and/or contributing to aging, associated morbidity and increased mortality [16]. It has been hypothesized that iron dyshomeostasis and ferroptosis are central to the mechanisms underlying the gradual dysfunction observed and consequently aging, certainly acting towards the end of life as “executioners” (vide infra).

Authors such as Stroustrup et al. [17] have shown that it is possible to affect longevity by interfering with several metabolic pathways such as insulin-like growth factor-1 (IGF-1), hypoxia-inducible factor (hif-1) and heat shock factor (HSF-1) on *C. elegans*, which alter lifespan with an apparent “time scaling”. It was found that salicylaldehyde isonicotinoyl hydrazone (SIH), a lipophilic acyl-hydrazone that eliminates intracellular iron for extracellular clearance, and liproxstatin-1 (Lip-1), an inhibitor of ferroptosis through the inactivation of lipid peroxide radicals, can reshape life stages. SIH reduces the risk of death in middle age through cellular downsizing, while Lip-1 acts in late post-reproductive age. While iron accumulation contributes to many processes that can modulate the rate of aging, inhibition of ferroptosis reduces frailty (late-life survival) rather than modulating the overall rate of aging. It is then possible that iron dyshomeostasis and ferroptosis are not central to the mechanisms underlying the observed gradual dysfunction but act towards the end of life as “executioners” to rapidly damage and degrade functions.

In a world where the average age of the population is steadily increasing, it is important to reduce the signs of senescence and age-related conditions in order to improve the quality of life of the elderly population and reduce healthcare costs worldwide. It is recommended by the WHO to explore and study the mechanisms that lead to this physiological decline, to prevent or treat age-related diseases by blocking or controlling the biological processes involved [18]. Ferroptosis is certainly one of these and must be properly studied and understood in order to implement effective strategies to counteract it or to exploit its tendency to eliminate senescent or damaged cells [18].

The aim of this study was to investigate the ferroptosis pathway activation and look at the features of iron-induced stress to understand the mechanisms correlating ferroptosis with senescence. Moreover, it was important to understand whether ferroptosis could be a new mechanism of senolytic therapy applicable for health longevity.

## 2. Methods

This review work was undertaken under the guidance of Preferred Reporting Items for Systematic Review and Meta-analysis.

### 2.1. Search Strategy

A comprehensive search of multiple databases was conducted in order to identify relevant studies. The databases searched were PubMed, Scopus, Google Scholar, Web of Science, and the Cochrane Library. The search included papers published until December 2022. The search terms used were “senescence”, “ferroptosis”, “aging” and “disease”. The search was limited to English language papers only.

### 2.2. Inclusion Criteria

This review considered studies investigating senescence and/or ferroptosis in any tissue or organ as they satisfied the eligibility criteria. Inclusion criteria were directed towards studies focusing on both human or animal subjects.

The following information was extracted from each paper: study design, sample size, participant characteristics such as age, gender, disease status, interventions, outcomes, and results. The study characteristics were then tabulated.

### 2.3. Quality Assessment

The quality of the included studies was assessed using the Cochrane Risk of Bias tool. This tool comprises of six domains: random sequence generation, allocation concealment, blinding participants and personnel, blinding of outcome assessment, incomplete outcome data, and selective reporting. Each domain was rated as “low risk”, “high risk”, or “unclear risk” based on the information provided in the study. Two independent reviewers performed the quality assessment, and any discrepancies were resolved through discussion and consensus. The overall quality of each study was rated as “low risk”, “high risk”, or “unclear risk” based on the ratings of the individual domains.

### 2.4. Analysis

This review aims to assess the current evidence on the potential mechanisms connecting senescence, ferroptosis, aging, and disease, although the exact pathways underlying this connection are still not fully understood.

The search of study selection identified a total of 982 studies that met the inclusion criteria. They comprehended a variety of designs, such as randomized controlled trials (RCTs), observational and in vitro studies. Most of the studies were conducted in animal models, although a few were in human subjects. Out of the 982 articles, 196 were excluded for being duplicates. Abstract screening resulted in the elimination of 181 more studies. Of the remaining 605 articles, 121 could not be retrieved. The study design screening further excluded 183 articles since they were deemed unsuitable. A further exclusion decision eliminated 196 more studies since they failed to report on the relationship linking senescence, ferroptosis, disease, aging, and cell death. Of the remaining papers, 71 were eliminated due to insufficient data, while some others were written in languages other than English. Therefore, 34 studies were deemed suitable for this review, as reported in Figure 2.

## 3. Results and Discussion

The involvement of ferroptosis in senescence development and progress can be inferred by the interesting results reported in a number of 34 studies focusing on the role of this regulated form of cell death (RCD) in aging-related processes, such as cardiovascular conditions, neurodegeneration, and cancer.

Zhou et al. (2020) explored the mechanisms of ferroptosis and the implications of aging and age-related complications. This is a particularly burdensome problem for healthcare systems worldwide given the rapidly increasing aging population. Researchers have concluded that ferroptosis contributes significantly to age-related disorders, including cancer, cardiovascular diseases, and neurodegenerative conditions. This assertion is related to the mechanism by which cells undergoing ferroptosis secrete factors that strongly activate the innate immune system. The authors point out that to date there is still a lack of clear evidence of ferroptosis in human cells and tissues in autoptic examinations, to understand the mechanism by which ferroptosis regulates degeneration, the main cause of tissue damage and organ failure [19].

Zhao, T. (2021) [20] discussed an overview of the mechanisms of iron accumulation and lipid peroxidation in the aging retina and the role of iron accumulation in the development of age-related macular degeneration (AMD), and the potential involvement of ferroptosis in this process. Intracellular iron concentration is finely regulated at three levels: control of iron uptake, modulation of the labile iron pool, and regulation of iron export [21,22,23,24]. Excess iron can be toxic by forming ROS through the Fenton’s mechanism, initiated by a reaction between ferrous iron and hydrogen peroxide [25]. Among the proteins involved in the phototransduction cascade in the retina [26], the RPE65 protein converts 11-cis-retinal to all-trans-retinyl as a part of the retinoid cycle required for iron-dependent phototransduction [27]. Furthermore, iron is essential for guanylate cyclase in the phototransduction pathway [26], but if this metal accumulates, it becomes toxic by forming ROS, exceeding the cellular antioxidant capacity. Thus, the aberrant production of ROS damages DNA and proteins and lipids within the retina [28,29]. This process leads to the production of two major products of lipid peroxidation, malondialdehyde (MDA) and 4-hydroxynonenal (4-HNE), which increase in the inner segments of photoreceptors, producing oxidative stress-induced cell death as the final event in the cell death cascade that underlies the pathogenesis of AMD. These findings have been observed both in AMD patients and animal models [30,31].

This, on the other hand, had already been highlighted by Yun Sun et al. in 2018 [32], when examining cultured ARPE-19 cells and the effects of GSH depletion on stress-induced premature senescence (SIPS). Cell death was caused by GSH depletion which in turn induced ferroptosis, but also autophagy and SIPS, with autophagy being a negative regulator of SIPS. However, the study did not clarify whether ferroptosis is a process of autophagy activation or a prerequisite for it.

Guo et al., (2022) [33] have carefully explored the molecular mechanisms of aging and age-related diseases and their respective treatments and interventions. The study of the mechanisms of aging included various areas such as epigenetic regulation, proteostasis, autophagy, cellular senescence, and stem cells. Among the many causes of aging, whose mechanisms are extremely complex, they highlighted how the levels of autophagy-related proteins directly influence the organismal aging. In other words, the study found that the concentration of autophagy-related proteins decreases in an individual with advancing age and that translocation to lysosomes is reduced, which promotes the aging of the organism. Furthermore, the researchers found much higher iron levels in senescent cells than in non-senescent or immortalized cells due to defective autophagic degradation of ferritin in lysosomes [33].

Evidence from other studies suggests that several mechanisms may be involved in the association of senescence, ferroptosis, aging, and disease [34]. Some of these mechanisms we already mentioned, such as lipid peroxidation, glutathione-peroxidase-4 enzyme activity, and iron metabolism, are able to trigger the secretion of pro-inflammatory cytokines and other molecules, including interleukin-6 (IL-6) and tumour necrosis factor-alpha (TNF-alpha) by senescent cells. Their involvement is relevant in the pathophysiology of certain neurodegenerative conditions such as Alzheimer’s, Parkinson’s, and Huntington’s diseases [35]. However, from a mechanistic point of view, ferroptosis is believed to be linked to the activation of these pro-inflammatory pathways, which may contribute to the death of damaged or sick cells and thus counteract senescence, characterized by the evasion of programmed cell death pathways.

It is worth noting that disease mechanisms are broad and vary depending on the type and changes involved at the organismal, cellular, or molecular level [36]. Some notable disease mechanisms are genetic mutations, environmental exposures, and infections. Mutations in specific genes can trigger changes in molecular structures and the production of abnormal proteins, resulting in disease [37,38]. In addition, exposure to environmental polyforms and other harmful substances can cause incidences such as cancer. One limitation of the studies included in this review is that many were conducted in animal models, which may not fully reflect the processes occurring in humans. In addition, the studies, reported in Table 1, included a variety of designs and methods, which may make it difficult to compare the results across studies. It is also worth noting that the mechanisms underlying the link among senescence, ferroptosis, aging, and disease are likely complex and multifaceted. They may involve other factors in addition to those discussed in this review. Further research is, however, needed to fully understand the role of these mechanisms and to identify potential interventions that could target such processes in order to prevent or treat these diseases.

From the observation of these results, it appears that senescence and ferroptosis may be directly related, but there are also many other studies that focused on specific aspects linked to the general decay of the organism during the aging process that should be examined in detail, as follows.

### 3.1. Brain Damage

Among age-related conditions, cerebrovascular diseases are some of the most frequent medical challenges raising awareness on the limitations in clinical treatment strategies. They represent the second leading cause of death in the world population and the sixth most common cause of disability. Ferroptosis, caused by abnormal metabolism of lipids, in which the brain is particularly rich, accelerates acute injury to the central nervous system. Recent studies have gradually uncovered the pathological process of ferroptosis in the neurovascular unit of acute stroke. Altered homeostasis of iron [54], the most abundant metal in the brain necessary for cerebral metabolism [46], produces abnormal overload in specific regions of the brain, promoting disease progression [37]. This process accompanies pathological aggravation and is considered to be an important pathophysiological factor involved in secondary damage after ischemic stroke [55]. The study aimed at evaluating molecular targets for plant active ingredients of natural products that regulate ferroptosis after ischemic stroke. The molecules evaluated belonged to the macro-families of saponins, flavonoids, and polyphenols. These biomolecules were used in animal models while a follow-up of clinical studies in large-scale, long-term patients is lacking.

Yihang Pan [45] put attention in describing the link between ferroptosis and ischemia-reperfusion (I/R) injury in various conditions, including ischemic stroke, myocardial infarction, heart attack, acute lung injury, liver transplantation, acute kidney injury, and haemorrhagic shock. Through the study of ferroptosis-regulated genes investigated in the context of I/R, they suggested beneficial applications of ferroptosis regulators as a therapeutic target to alleviate I/R injury. Other authors described the pathological process of I/R to explore the molecular basis in the pathogenesis of ferroptosis and evaluate its role. They also analysed the role of tested iron-chelating agents that have shown efficacy in improving outcomes in a variety of I/R-associated symptoms, while not overlooking the potentially negative impact on the bloodstream and little organ specificity, suggesting that iron alone may not be an optimal target [56].

It was previously reminded that iron dyshomeostasis and lipid peroxidation are associated with aging. Iron homeostasis in brain is finely regulated by a delicate balance of iron movement between blood and tissue across the blood-brain barrier, intracellular and extracellular environments, and between different iron pools. Imbalances in these processes are the major risk factor for neurodegenerative diseases such as Alzheimer’s disease (AD) and Parkinson’s disease (PD) where intracellular iron accumulation was first observed. For this reason, a number of studies and reviews attempted to elucidate the role of ferroptosis in the brain and the understanding of the mechanism of neurological diseases to provide potential prevention and treatment interventions for neurological conditions including neurodegeneration, stroke, and neurotrauma [48]. Masaldan et al., (2018) have studied senescence in mouse embryonic fibroblasts (MEFs) and measured intracellular iron by inductively coupled plasma mass spectrometry (ICP-MS). While examining the relationship between altered cellular iron acquisition and storage, they noted a link between impaired ferritinophagy, a lysosomal process that explains the accumulation of iron, and ferroptosis development [39]. Thereafter, the same research group (2019), gathered evidence linking neurodegenerative disorders to the induction of senescence in brain cells following iron accumulation while examining iron dyshomeostasis in the context of Alzheimer’s disease [40].

Derry et al. as well analysed the dysfunctions involved in the course of Alzheimer’s disease, among all the pathologies associated with senescence. Increased sensitivity to ferroptosis is related to the numerous binding mechanisms of tau with the various forms of iron, which leads to its accumulation in the brains of individuals with neurodegenerative disorders.

Furthermore, it should be added that in AD certain aspects of the glutathione synthesis and utilization pathways are disrupted. Analysis of the mechanisms led to the proposal of new pharmaceutical targets and therapies that address multiple pathways, in particular the iron-mediated ones [52].

Consistent with this study is that of Majerníková, den Dunnen and Dolga (2021), where the authors evaluated the potential of therapeutic interventions on ferroptosis as a treatment for AD. The authors confirmed that ferroptosis contributes to AD development and progression. The study also explored other mechanisms, such as transcriptomic analysis, to help understand the impacts of ferroptosis on Alzheimer’s disease targeted therapies [41].

Altered interaction in the metabolic and nutritional coupling between glial cells (astroglial cells, oligodendrocytes, and microglia) and neurons can lead to neuronal death due to excessive iron accumulation, especially in ferroptosis [47]. In the central nervous system, iron, as an important cofactor [57], is mainly combined with ferritin and neuromelanin and participates in several important processes, including oxygen transport, oxidative phosphorylation, myelin production and neurotransmitter synthesis and metabolism [58].

Ren et al., compared the differences and relationships among the various cell death mechanisms involved in neurological diseases to elucidate the role of ferroptosis. This could improve the understanding of the mechanism to provide potential treatments for acute and chronic neurological disorders [48].

Among all the dysfunctions threatening human health and quality of life, Wang et al. (2022) [49], focused their attention on the emerging role of ferroptosis in cardiovascular conditions as a potential therapeutic target. In recent years, the pathogenic role of iron overload in cardiotoxicity has been recognized, mainly on animal models and cellular levels. [59] found that protein p53 participates in the nonclassical pathway of ferroptosis regulation. If better understood, this pathway could be a valuable tool in the prevention and treatment of cardiovascular disease, but experimental verification is still lacking in vivo.

### 3.2. Cancer

Ferroptosis is implicated in several pathological pathways, as suggested by Yang et al. [46] in a review where they summarize the current knowledge on the mechanism of iron regulation, and lipid and cysteine metabolism. They further discussed the contribution of ferroptosis to the pathogenesis of several diseases. It emerged that ferroptosis may be useful in circumventing therapy resistance of cancer cells especially after they have acquired drug resistance. Neutrophils can promote, at the tumour site, myeloperoxidase (MPO), which in turn catalyses iron-dependent phospholipid peroxidation and promotes ferroptosis in neighbouring tumour cells, showing a strong anti-tumour function [60]. In addition, ferroptosis may participate in cancer immunity through adaptive mechanisms. The role of ferroptosis in cancer is therefore ambiguous, as it may be either an immunogenic or immunosuppressive form of cell death [61]. Therefore, it may represent an exploitable feature critical for the treatment of cancer and certain types of diseases. In fact, understanding the mechanisms of a form of regulated cell death driven by phospholipid peroxidation could prove to be a promising approach in the treatment of a range of conditions, including neoplasms. The growing body of research has allowed describing at least three pathways controlling the sensitivity of cells to ferroptosis: glutathione-GPX4, NADPH-FSP1-CoQ 10 and GCH1-BH4 pathways. To date, it is known that ferroptosis is the most conserved form of cell death in all the diversity of life on earth than any other, and cells have evolved a complex control system to regulate when and how to activate ferroptosis and prevent tumorigenesis. Identification of targets to induce and inhibit ferroptosis is a promising strategy for future therapies [1] since, for instance, it activates the NRF2 pathway, altering the associated molecular makeup [62]. Dixon and Stockwell (2019) linked ferroptosis to the function of key tumour suppressor pathways, underlining how this is an interesting checkpoint in stress responses, and in tumour suppressor mechanisms. Indeed, protein p53 is known to affect cell cycle arrest, senescence, and apoptosis. However, its interaction with ferroptosis is of great interest [8]. Through its metabolic targets, p53 modulates the response to ferroptosis via the canonical (GPX4-dependent) and non-canonical (GPX4-independent) ferroptosis pathways [63]. Among transcriptional target of p53 some are involved in polyamine catabolism, molecules known to be involved in tumorgenesis. Activation of the expression of spermidine/spermine N1-acetyltransferase 1 (SAT1), through lipid peroxidation is involved in sensitisation of cells to ferroptosis [64,65,66,67,68].

Tong et al., described in a review the mechanisms of necroptosis, pyroptosis, ferroptosis, and cuproptosis and the effects on tumour cell proliferation and cancer metastasis. The authors draw attention onto the potential agents and nanoparticles for cancer treatment or the association of these with already existing therapies inducing RCD compared to conventional treatments [50].

Gong et al. (2022) explored the role of regulated cell death on cancer development and potential interventions and treatments [42]. The authors examined how ferroptosis contributes to the development and progression of cancer in the aging population through the reduction in mitochondria via parkin-mediated mitophagy which significantly reduces the susceptibility of cells to ferroptosis induced by cysteine deprivation, as already observed in the literature.

Another paper by Gao et al. also established the crucial role of mitochondria in ferroptosis. Cysteine deprivation leads to hyperpolarisation of the mitochondrial membrane and accumulation of lipid peroxide. The complex mechanism discussed by the authors implies the relevance of ferroptosis in tumour suppression [51].

These findings led Bano and colleagues to define ferroptosis as a new avenue for cancer management after analysing the factors and molecular mechanisms playing a role in the initiation and susceptibility of ferroptosis in various malignancies [69].

### 3.3. Further Findings

Most of the studies showed that ferroptosis has a pathological role in various contexts. A very interesting finding in this area of research is microbial virulence factors’ alleged ability to exploit or dampen ferroptosis regulatory pathways to their own benefit and the immune consequences. During an infection, reactive oxygen species are rapidly increased to facilitate pathogen elimination and implement inflammation and immune responses [70].

There is indirect evidence linking lipid peroxidation, iron, and ferroptosis to host-pathogen interactions whether they are bacteria, viruses, fungi, or parasites. Attention is predominantly focused on the involvement of lipid peroxidation-driven ferroptosis in infectious diseases through its ability to spread a ferroptosis signal to neighbouring cells following a “wave” pattern [53]. This paradigm will probably help translate ferroptosis to various clinical contexts.

One limitation of the studies included in this review is that many of them were conducted in animal models, which may not fully reflect the processes occurring in humans. In addition, the studies included a variety of designs and methods, which may make it difficult to compare the results across studies. It is also worth noting that the mechanisms underlying the link connecting senescence, ferroptosis, aging, and disease are likely complex and multifaceted. They may involve other factors in addition to those discussed in this review. Further research is, however, needed to fully understand the role of these mechanisms, as in Figure 1 are described, to identify potential interventions that could target these mechanisms to prevent or treat age-related diseases.

Ferroptosis pursues upon build-up of lipid reactive oxygen species (ROS) leading to peroxidation of polyunsaturated fatty acids (PUFAs). Lipid peroxidation can be triggered by cytosolic redox active iron (Fe^2+^) shuttled into cells bound to transferrin via transferrin receptor (TFRC) endocytosis and endosomal release mediated by divalent metal transporter 1 (DMT1). In the presence of H_2_O_2_, Fe^2+^ catalyses hydroxyl radical (HO∙) generation in Fenton reaction, which sets of a radical lipid peroxidation chain reaction. Lipoxygenase (LOX) can equally catalyse lipid peroxidation using Fe^2+^. Glutathione peroxidase 4 (GPX4), in turn, hydrolyses lipid peroxides converting them into their respective non-toxic lipid alcohols (-OH). GPX4 requires glutathione (GSH) as a cofactor which upon its oxidation (GSSG) by GPX4 is reduced to GSH by glutathione reductase (GR). GSH synthesis depends on glutamate cysteine ligase (GCL) and glutathione synthetase (GSS) as well as on intracellular cystine shuttled into the cell in exchange for glutamate mediated by system xc-(SLC3A2 and SCL7A11/xCT).

## 4. Conclusions

In its design, this review has been limited to research papers that uncovered the complex connection between senescence and ferroptosis, which presents itself as a new field with great potential in helping people age in good health, by expanding medical indications and promoting the clinical exploitation of ferroptosis to improve the life quality for senior people. In this perspective, further trials are urgently needed to improve our knowledge on the underlying mechanisms, in order to be able to translate the new findings into medical therapy and intervention. In fact, ferroptosis has been described in several studies that investigated its initial and intermediate signals and pathways, but true biomarkers that can be easily correlated to this process are still unknown. In actuality, the biomarkers of ferroptosis analysed so far are also common to other mechanisms and pathways. The use of new technologies could help discover more specific biomarkers, the “performers”, which could provide new opportunities for designing new treatments for iron overload and age-related diseases. The challenge is to turn basic research into clinical applications.

As for the present, the data reviewed so far confirm that several mechanisms may be interrelated and involved in the link connecting senescence, ferroptosis, aging, and disease. These mechanisms include the production of pro-inflammatory cytokines and other molecules by senescent cells, the accumulation of ROS and other oxidative stress-related species, and the activation of cell death pathways. Ferroptosis appears to play a role in various age-related diseases, including diabetic nephropathy, cardiovascular diseases, ischemia/reperfusion-related injury, and age-related macular degeneration. Some studies also suggest that ferroptosis may be a potential therapeutic target to counteract senescence, while others provide an overview of the mechanisms and links between ferroptosis and the abovementioned diseases. Overall, they indicate that ferroptosis may be a promising area of research for understanding and potentially treating age-related conditions. They also reveal many research gaps in understanding the role of ferroptosis and its potential mechanisms connected with senescence, together with the need for large-scale clinical studies, as most of the available literature is relatively limited, with only a few dozen participants, to confirm these studies’ findings and determine the generalizability of the results for future research in these areas.

## Data Availability

Not applicable.

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
