# Peer review of "Ferroptosis and Senescence: A Systematic Review"

_ijms, 2023, doi:10.3390/ijms24043658_

Round 1

Reviewer 1 Report

Suggestion: Please proofread an y grammatical errors and typos. 

One more point: Is there any way the reports excluded from Abstract selection contain some vital information. Just making sure exclusion with some keywords is not causing loss of data. Maybe this can be revisited.

I like the conclusion overall and mainly the discussion of limitations of the study. Maybe one or two lines describing an ideal way of experimentation in your opinion would be a good addition.

Author Response

Dear Reviewer,

We truly appreciate the thoughtful comments provided by the Editor and the Reviewers, and would like to thank them for helping us improve the quality of our work. We have extensively revised the manuscript accordingly and report our point-to-point answers below.

We are additionally incorporating a new figure for a better illustration of the concepts.

Yours sincerely,

Ciriaco Carru

Suggestion: Please proofread any grammatical errors and typos. 

This has been done for the whole text.

One more point: Is there any way the reports excluded from Abstract selection contain some vital information. Just making sure exclusion with some keywords is not causing loss of data. Maybe this can be revisited.

We carried out two independent searches on different databases and examined also papers that did not strictly meet the inclusion criteria. It seemed that they did not add new information on the topic or were not eligible for inclusion, so we just kept those emerged from the original selection.

I like the conclusion overall and mainly the discussion of limitations of the study. Maybe one or two lines describing an ideal way of experimentation in your opinion would be a good addition.

Thank you very much. Another reviewer suggested some more changes in the conclusions, so we tried to merge the two indications in the best way we could.

Reviewer 2 Report

1. Abstract --> abstract should no provide citation notation

2. Introduction :

a. Authors must declare the gap or urgency of study. 

b. Authors should declare the aim of study.

3.Methods: 

a. PRISMA diagram should be revised with "arrow"

4. Discussion:

a. Authors should decribe the pathway, how feroptosis can induce apoptosis, inflammation and cancer?

5. Conclusion:

a. Authors should provide the conclusion more simple and conclude the results of study.

Author Response

Dear Reviewer, 

We truly appreciate the thoughtful comments provided by the Reviewers and would like to thank them for helping us improve the quality of our work. We have extensively revised the manuscript accordingly and report our point-to-point answers below.

We are additionally incorporating a new figure for a better illustration of the concepts.

Yours sincerely,

Ciriaco Carru

  1. Abstract --> abstract should no provide citation notation

Citation in the abstract has been eliminated

  1. Introduction:

a.Authors must declare the gap or urgency of study. 

b.Authors should declare the aim of study.

These two points have been addressed and improved in the introduction

3.Methods: 

a. PRISMA diagram should be revised with "arrow"

We have revised the diagram accordingly

  1. Discussion:

a. Authors should describe the pathway, how ferroptosis can induce apoptosis, inflammation and cancer?

Ferroptosis is a unique type of non-apoptotic regulated cell death (RCD), so it does not induce apoptosis. Several studies have indicated that known cancer suppressors are involved in the modulation of ferroptosis such as p53, a cancer suppressor that mediates cell cycle arrest, senescence and plays a role in the regulation of ferroptosis. Some references can be found for instance at: Li T., Kon N., Jiang L., Tan M., Ludwig T., Zhao Y., et al. (2012). Tumour suppression in the absence of P53-mediated cell cycle arrest, apoptosis and senescence (Cell 149, 1269-1283. 10.1016/j.cell.2012.04.026). Activation of the expression of spermidine/spermine N1-acetyltransferase 1 (SAT1), a transcriptional target of p53 involved in polyamine catabolism, induces through lipid peroxidation the sensitisation of cells to ferroptosis (Ou Y., Wang S.-J., Li D., Chu B., Gu W. (2016). SAT1 activation involves polyamine metabolism with P53-mediated ferroptotic responses. Proc. Natl. Acad. Sci. USA 113, E6806-E6812. 10.1073/pnas.1607152113). Another tumour suppressor, breast cancer type 1 (BRCA1)-associated protein 1 (BAP1), has also been explored for its role in ferroptosis (Zhang Y., Shi J., Liu X., Feng L., Gong Z., Koppula P., et al. (2018). BAP1 links metabolic regulation of ferroptosis to tumor suppression. Nat. Cel Biol 20, 1181-1192. 10.1038/s41556-018-0178-0).

We think we have discussed the mechanisms involved in the induction of inflammation and cancer by ferroptosis with enough details throughout the dedicated sections in the manuscript, but to improve clarity we added a new figure summarizing these aspects.

  1. Conclusion:

a.  Authors should provide the conclusion more simple and conclude the results of study.

We have revised the conclusions according to the suggestions of two reviewers, we hope we found a good synthesis of the two indications.

Reviewer 3 Report

Dear Authors

I have reviewed your enthusiastic review manuscript entitled “Ferroptosis and Senescence: a systematic review”, the manuscript is well-written and for more understandable, but it should be revised as follow:

1. I strongly recommend the authors to revise the English writing of the manuscript, mainly, in the case of word and collocations, i.e the word of decline (line 12), illness (line 17) used instead of disease, and so on.

2. The figures are of low quality, which should be enhanced.

3. Line 24, can you explain why the citation is applied in the abstract, while this is not popular?

4. Lines 24-27 are not comprehensible, rewrite them.

5.   As a review paper, the text is poor in the case of figures, graphical abstracts, images, and so on, I strongly recommend inserting more figures and images, due to the impact of images being stronger than text.

Overall, although the topic and contents are both of interest, the manuscript should be rewritten in English into more comprehensible text.

Author Response

Dear Reviewer,

We truly appreciate the thoughtful comments provided by the Reviewers and would like to thank them for helping us improve the quality of our work. We have extensively revised the manuscript accordingly and report our point-to-point answers below.

We are additionally incorporating a new figure for a better illustration of the concepts.

Yours sincerely,

Ciriaco Carru

Reviewer 3

I have reviewed your enthusiastic review manuscript entitled “Ferroptosis and Senescence: a systematic review”, the manuscript is well-written and for more understandable, but it should be revised as follow:

  1. I strongly recommend the authors to revise the English writing of the manuscript, mainly, in the case of word and collocations, i.e the word of decline (line 12), illness (line 17) used instead of disease, and so on.

We have extensively revised the manuscript for English writing and typos

  1. The figures are of low quality, which should be enhanced.

The quality of the figures has been enhanced

  1. Line 24, can you explain why the citation is applied in the abstract, while this is not popular?

Citation in the abstract has been eliminated

  1. Lines 24-27 are not comprehensible, rewrite them.

Lines 24-27 have been rewritten

  1.   As a review paper, the text is poor in the case of figures, graphical abstracts, images, and so on, I strongly recommend inserting more figures and images, due to the impact of images being stronger than text.

Graphical abstract and figure 1 have been added

Overall, although the topic and contents are both of interest, the manuscript should be rewritten in English into more comprehensible text.

The whole manuscript has been revised according to this suggestion

Round 2

Reviewer 3 Report

Dear Authors,

Thanks your revised manuscript, now it is acceptable,

just as a suggestion for your figures, for example figure 1, you can design your figure in power point (ppt), then press Control +A, Copy it and then past as a picture in word MS and finally resize it

BR